# Do sex differences in the prevalence of ECG abnormalities vary across ethnic groups living in the Netherlands? A cross-sectional analysis of the population-based HELIUS study

Renee Bolijn [ID],[1] C Cato ter Haar,[2,3] Ralf E Harskamp,[4] Hanno L Tan,[2,5] Jan A Kors,[6] Pieter G Postema,[7] Marieke B Snijder,[1,8] Ron J G Peters,[2] Anton E Kunst,[1] Irene G M van Valkengoed[1]

For numbered affiliations see end of article.

**Correspondence to**
Renee Bolijn;
r.bolijn@amsterdamumc.nl

## ABSTRACT

**Objectives** Major ECG abnormalities have been associated with increased risk of cardiovascular disease (CVD) burden in asymptomatic populations. However, sex differences in occurrence of major ECG abnormalities have been poorly studied, particularly across ethnic groups. The objectives were to investigate (1) sex differences in the prevalence of major and, as a secondary outcome, minor ECG abnormalities, (2) whether patterns of sex differences varied across ethnic groups, by age and (3) to what extent conventional cardiovascular risk factors contributed to observed sex differences.

**Design** Cross-sectional analysis of population-based study.

**Setting** Multi-ethnic, population-based Healthy Life in an Urban Setting cohort, Amsterdam, the Netherlands.

**Participants** 8089 men and 11 369 women of Dutch, South-Asian Surinamese, African Surinamese, Ghanaian, Turkish and Moroccan origin aged 18–70 years without CVD.

**Outcome measures** Age-adjusted and multivariable logistic regression analyses were performed to study sex differences in prevalence of major and, as secondary outcome, minor ECG abnormalities in the overall population, across ethnic groups and by age-groups (18–35, 36–50 and >50 years).

**Results** Major and minor ECG abnormalities were less prevalent in women than men (4.6% vs 6.6% and 23.8% vs 39.8%, respectively). After adjustment for conventional risk factors, sex differences in major abnormalities were smaller in ethnic minority groups (OR ranged from 0.61 in Moroccans to 1.32 in South-Asian Surinamese) than in the Dutch (OR 0.49; 95% CI 0.36 to 0.65). Only in South-Asian Surinamese, women did not have a lower odds than men (OR 1.32; 95% CI 0.96 to 1.84). The pattern of smaller sex differences in ethnic minority groups was more pronounced in older than in younger age-groups.

**Conclusions** The prevalence of major ECG abnormalities was lower in women than men. However, sex differences were less apparent in ethnic minority groups. Conventional risk factors did not contribute substantially to observed sex differences.

## Strengths and limitations of this study

► Participants were sampled from the municipality registry and reflect a general population sample of adults of the major ethnic groups living in Amsterdam.

► Large sample sizes permit the study of sex differences within each ethnic group, including across age strata.

► Single ECG measurements of 10 s without additional imaging techniques (eg, echocardiography) may be suboptimal for the measurement of ECG abnormalities.

► The classification of 'major' and 'minor' ECG abnormalities may depend on criteria used, which may affect reported prevalence estimates.

## INTRODUCTION

The resting ECG is an essential diagnostic instrument in patients with symptoms suggestive of cardiovascular disease (CVD).[1] Previous studies show that the occurrence of major ECG abnormalities is also associated with increased risk of CVD morbidity[2 3] and mortality[3 4] in asymptomatic populations. However, differences in the occurrence of major ECG abnormalities in men and women have been poorly studied. Insights in these sex differences may help to identify subpopulations with a future CVD burden and thus aid targeted (preventive) therapy.

Although studies have described the prevalence of major ECG abnormalities in men and women from diverse general populations,[5–11] only three studied sex differences in general populations specifically.[5–7] Whether the occurrence of ECG abnormalities differs by sex, independently of cardiovascular risk factors, is a topic of ongoing debate. Two

studies suggested that the composite of major ECG abnormalities (eg, atrial fibrillation, Q-wave or T-wave abnormalities) is more prevalent among men than women,[5 6] while another observed no sex differences.[7]

Differences in the occurrence of ECG abnormalities have been observed between ethnic groups living in similar contexts.[6 8 12 13] However, in Europe, the prevalence of major ECG abnormalities among ethnic minority populations at high risk for CVD, such as men and women of South-Asian origin,[14] is unknown. Additionally, it is unknown to what extent major ECG abnormalities vary between men and women across ethnic groups.

The prevalence of ECG abnormalities tends to increase with increasing age.[5] As larger sex differences in occurrence of CVD have been found in younger age-groups compared with older age-groups,[15] sex differences in prevalence of ECG abnormalities may also vary by age.

In this study, we investigated sex differences in the prevalence of major and, as a secondary outcome, minor ECG abnormalities, in a multi-ethnic population aged 18–70 years living in Amsterdam, the Netherlands. We assessed whether patterns of sex differences varied across ethnic groups, overall and by age, and to what extent conventional cardiovascular risk factors contributed to observed sex differences, overall and within subgroups.

## METHODS

We used baseline data from the Healthy Life in an Urban Setting (HELIUS) study, a multi-ethnic cohort study conducted in Amsterdam, the Netherlands. The HELIUS study has been described in detail elsewhere.[16 17] Briefly, baseline data collection took place between 2011 and 2015 and included participants of Dutch, South-Asian Surinamese, African Surinamese, Ghanaian, Moroccan and Turkish origin aged 18–70 years living in Amsterdam. Potential participants were sampled with a simple random sampling method from the municipality registry, after stratification by ethnicity as defined by registered country of birth.[18] Data were obtained by questionnaire and physical examinations (including biological samples). The HELIUS study has been approved by the AMC Ethical Review Board. All participants provided written informed consent.

### ECG measurements

Standard 12-lead ECGs were recorded in supine position with a GE MAC5500 ECG at 500 samples/s and analysed using the Modular ECG Analysis System (MEANS).[19] The measurement of ECG abnormalities has been described in detail elsewhere.[20] Briefly, ECG abnormalities were assessed by combining ECG diagnoses of the MEANS programmes with Minnesota coding, Marquette 12SL ECG analysis software and a cardiologist's interpretation. In case of discrepancies, ECGs were double checked. We classified ECG abnormalities into major and minor ECG abnormalities, based on previous research[7] and consensus discussion among experts (online supplementarhy appendix table 1). This classification was completed prior to data analysis.

### Ethnicity

Ethnicity was defined by the individual's country of birth combined with the parental countries of birth.[18] Surinamese participants were further classified according to self-reported ethnic origin into 'African', 'South-Asian', 'Javanese' or 'other'.

### Covariables

Family history of CVD was defined by a self-reported CVD diagnosis among first degree family members aged <60 years. Smoking was classified as current, past or never smoker. For current smokers, the number of pack-years of smoking was calculated by multiplying the number of packs (containing 20 cigarettes or equivalent rates for cigars and pipe tobacco) smoked a day by the number of years smoked. Physical activity was defined as achieving ≥30 min of moderate-intensity or high-intensity activity per day on ≥5 days per week.[21] Alcohol consumption (on average in the last 12 months) was classified as: none or low (men: 0–4; women: 0–2 beverages/week), moderate (men: 5–14; women: 3–7 beverages/week) and high (men: >14; women: >7 beverages/week).

Body mass index (BMI) was calculated in duplicate as weight (kg) divided by height squared ($m^2$). Blood pressure (BP) was measured in duplicate using a validated automated digital BP device (WatchBP Home; Microlife AG) in a seated position after ≥5 min of rest. Hypertension was defined as systolic BP ≥140 mm Hg, diastolic BP ≥90 mm Hg, use of antihypertensive medication treatment and/or self-reported hypertension.

Fasting blood samples were drawn to determine creatinine, lipid and glucose concentrations (details on these measurements have been described elsewhere).[22] Chronic kidney disease (CKD) risk was categorised according to the risk of progression of kidney disease based on estimated glomerular filtration rate and albuminuria levels[23]: (1) low, (2) moderately increased and (3) high and very high risk. Hypercholesterolemia was defined as total cholesterol ≥5.0 mmol/L, high-density lipoprotein cholesterol <1.0 mmol/L (men) or <1.2 mmol/L (women), low-density lipoprotein cholesterol ≥3.0 mmol/L (Friedewald formula[24]), triglycerides ≥1.7 mmol/L, use of lipid-lowering medication and/or self-reported hypercholesterolemia. Participants were considered to have diabetes in case of a fasting glucose ≥7.0 mmol/L, use of glucose-lowering medication and/or if they reported to be diagnosed with diabetes by a doctor.

### Study population

Baseline data were available for 22 165 participants. We excluded those of Javanese Surinamese (n=233), unknown Surinamese (n=267) origin and with another/unknown ethnic origin (n=48). Next, we excluded participants with a history of CVD (n=1610; based on self-reported prior myocardial infarction, cerebrovascular accident (CVA),

angioplasty or bypass surgery (on heart or legs), use of antiplatelet drugs (Anatomical Therapeutic Chemical (ATC) code B01AC), use of oral anticoagulants (ATC codes B01AA, B01AE, B01AF), use of antiarrhythmic agents (ATC codes C01A, C01B, C07AA07, C08D), or paced rhythms). Finally, we excluded participants with missing ECG data (n=337) or with missing data on ≥1 covariables (n=212), resulting in a study population of 19 458 participants (online supplementary appendix figure 1).

## Statistical analyses

Baseline characteristics were expressed as means (SD) or frequencies (percentages) by sex in the total population and per ethnic group. The age-adjusted prevalence of any major ECG abnormality, any minor ECG abnormality and a selection of common ECG abnormalities (ie, major ECG abnormalities with a prevalence of ≥1% and the top 5 most prevalent minor ECG abnormalities) was calculated by sex, in the total population and by ethnicity, using the study population as the standard. For reference, the overall prevalence of less common ECG abnormalities is also provided, but only by sex in the total population. The prevalence of any major ECG abnormalities was also calculated by age-groups (ie, 18–35, 36–50 and >50 years based on tertiles of the age distribution in the total population) for all ethnic groups.

We performed binary logistic regression analyses with hierarchal models to examine sex differences in prevalence of (1) any major ECG abnormalities and (2) any minor ECG abnormalities, adjusted for age and ethnicity (model 1), and additionally for hypertension, hypercholesterolemia, diabetes and smoking status (model 2) to determine to what extent conventional cardiovascular risk factors contributed to observed differences. We also examined the additional contribution of other well-known cardiovascular risk factors, that is, family history of CVD and CKD risk (model 3) and BMI, alcohol consumption and physical activity (model 4). To study whether the sex differences varied between ethnic groups (ie, effect modification), a statistical interaction term for sex and ethnicity on a multiplicative scale was added. Then, the main analyses (model 2 with interaction term) for major ECG abnormalities were repeated stratified by age-groups (18–35, 36–50 and >50 years) to examine the consistency of sex differences across ethnic groups among age-groups. All statistical analyses were performed in R studio V.1.1.453.[25] P values<0.05 were regarded as statistically significant.

## Sensitivity analyses

We repeated the main analyses excluding obese participants (BMI >30), since obesity may influence the accuracy of ECG measurements.[26] Furthermore, use of psychotropic medication may induce alterations of the ECG resulting in ECG abnormalities (eg, QT prolongation).[27] Therefore, we repeated the analyses excluding participants with current use of psychotropic medication.

Finally, we repeated the analyses using number of pack-years of smoking instead of smoking status, to examine whether the scale of the variables (numeric vs categorical) altered the results.

## Patient and public involvement

There was no specific patient or public involvement in the development of the research questions, outcome measures, study design and recruitment/conduct of the present study. However, for the core HELIUS study, several supportive measures were taken to enhance the enrolment of ethnic minority groups. For example, ethnic-specific communication strategies were used, such as working with faith communities (churches and mosques) and endorsement from local key figures. Understandability of and time to complete the questionnaire were also enquired among participants. In addition, the present study is part of a larger project on sex and gender differences in CVD risk. As part of this project, interviews and a short survey on research priority setting according to patients with CVD and persons at increased CVD risk were conducted. The present study aligns with our findings from these interviews and survey that more research on sex and gender differences in CVD was perceived as relevant by the target group.

## RESULTS

Mean age was around 43 years (SD 13) in women and 44 years (SD 13) in men (table 1). More than 20% of both men and women had a family history of CVD. Women were less often current smokers and had fewer mean pack-years of smoking compared with men, while the prevalence of high alcohol consumption was similar among men and women. Women had a higher mean BMI and were less physically active. Hypertension, hypercholesterolemia and diabetes were less prevalent among women than men, while high CKD risk was equally prevalent among men and women. Women more often used psychotropic medication than men. These patterns in baseline characteristics differed across ethnic groups (online supplementary appendix table 2).

Overall, the age-adjusted prevalence of major ECG abnormalities was lower among women (4.6%) compared with men (6.6%; table 2). In most ethnic groups, women had a lower age-adjusted prevalence (range: 2.9%–6.1%) compared with men (range: 4.7%–7.9%), except in the South-Asian Surinamese (7.2% vs 6.0%, respectively). Conventional cardiovascular risk factors and other well-known risk factors did not contribute substantially to the observed sex differences in major ECG abnormalities in the total population and within ethnic groups. For instance, the OR of having a major ECG abnormality changed from 0.69 (95% CI 0.61 to 0.78) to 0.71 (95% CI 0.62 to 0.81) among women vs men after adjustment for hypertension, hypercholesterolemia, diabetes and smoking status, and to 0.67 (95% CI 0.58 to 0.76) after

**Table 1** Baseline characteristics of 19 458 men and women with ECG measurements

| | Men (n=8089) | Women (n=11 369) |
|---|---|---|
| Age (years) | 43.8 (13.0) | 43.1 (13.0) |
| Ethnicity | | |
| Dutch | 1873 (23.2) | 2293 (20.2) |
| South-Asian Surinamese | 1125 (13.9) | 1464 (12.9) |
| African Surinamese | 1411 (17.4) | 2266 (19.9) |
| Ghanaian | 822 (10.2) | 1321 (11.6) |
| Turkish | 1451 (17.9) | 1769 (15.6) |
| Moroccan | 1407 (17.4) | 2256 (19.8) |
| Family history of CVD (missing: n=217) | 1637 (20.4) | 2611 (23.3) |
| Smoker | | |
| Current | 2539 (31.4) | 2032 (17.9) |
| Past | 2021 (25.0) | 1753 (15.4) |
| Never | 3529 (43.6) | 7584 (66.7) |
| Pack-years of smoking (missing: n=191) | 5.4 (16.2) | 1.8 (7.3) |
| Achieving physical activity norm (missing: n=27) | 5020 (62.2) | 5963 (52.5) |
| Alcohol consumption (missing: n=115) | | |
| None or low | 5981 (74.4) | 8985 (79.5) |
| Moderate | 1526 (19.0) | 1549 (13.7) |
| High | 528 (6.6) | 774 (6.8) |
| BMI (kg/m$^2$; missing: n=15) | 26.3 (4.2) | 27.5 (5.8) |
| CKD risk (missing: n=63) | | |
| Low | 7684 (95.4) | 10 689 (94.3) |
| Moderate | 304 (3.8) | 555 (4.9) |
| High | 68 (0.8) | 95 (0.8) |
| Hypertension | 3026 (37.4) | 3594 (31.6) |
| Hypercholesterolemia | 5752 (71.1) | 7147 (62.9) |
| Diabetes | 829 (10.2) | 952 (8.4) |
| Use of psychotropic medication (missing: n=4)* | 397 (4.9) | 679 (6.0) |

Data are presented as means (SD) or frequencies (%).
*Anatomical Therapeutic Chemical codes: N03AE, N03AF, N03AG, N03AN, N05A, N05BA, N05C, N06A, N06BA, N07B and R06AD02.
BMI, body mass index; CKD, chronic kidney disease; CVD, cardiovascular disease.

adjustment for family history of CVD, CKD risk, BMI, alcohol consumption and physical activity.

There was a general pattern of smaller sex differences in occurrence of major ECG abnormalities in the ethnic minority groups compared with the Dutch (table 2). The adjusted OR for women vs men varied from 0.49 (95% CI 0.36 to 0.65) in the Dutch to 0.73 (95% CI 0.53 to 1.01) in Turkish. Only in the South-Asian Surinamese group, women did not have a lower odds than men (adjusted OR 1.32; 95% CI 0.96 to 1.84).

In the total population, the most frequently observed major ECG abnormalities were T-wave abnormalities (1.2%), microvoltages (1.2%) and (ECG suggestive of)

**Table 2** Number of cases and age-adjusted prevalence of any major ECG abnormality by sex in the total population and by ethnic group, and the odds of major ECG abnormalities in women compared with men, overall and with an interaction term for sex and ethnicity

| | Men (no of cases, %*) | Women (no of cases, %*) | Model 1 | | | | Model 2 | | | |
|---|---|---|---|---|---|---|---|---|---|---|
| | | | OR (95% CI) | P value | Ratio of ORs (95% CI)‡ | P value | OR (95% CI) | P value | Ratio of ORs (95% CI)‡ | P value |
| Overall | 540 (6.6) | 518 (4.6) | 0.69 (0.61 to 0.78)† | <0.001 | NA | NA | 0.71 (0.62 to 0.81)† | <0.001 | NA | NA |
| Dutch | 137 (7.3) | 79 (3.6) | 0.46 (0.34 to 0.61) | <0.001 | Reference | NA | 0.49 (0.36 to 0.65) | <0.001 | Reference | NA |
| SA Surinamese | 63 (6.0) | 110 (7.2) | 1.24 (0.90 to 1.72) | 0.19 | 2.69 (1.75 to 4.17) | <0.001 | 1.32 (0.96 to 1.84) | 0.09 | 2.72 (1.76 to 4.21) | <0.001 |
| African Surinamese | 107 (7.6) | 118 (5.3) | 0.68 (0.51 to 0.89) | <0.01 | 1.47 (0.99 to 2.18) | 0.06 | 0.68 (0.52 to 0.90) | <0.01 | 1.40 (0.94 to 2.09) | 0.10 |
| Ghanaian | 72 (7.9) | 71 (6.1) | 0.70 (0.49 to 0.98) | 0.04 | 1.51 (0.97 to 2.37) | 0.07 | 0.71 (0.51 to 1.01) | 0.055 | 1.47 (0.94 to 2.30) | 0.09 |
| Turkish | 89 (6.1) | 77 (4.4) | 0.71 (0.52 to 0.98) | 0.04 | 1.55 (1.01 to 2.37) | 0.045 | 0.73 (0.53 to 1.01) | 0.058 | 1.51 (0.98 to 2.32) | 0.06 |
| Moroccan | 72 (4.7) | 63 (2.9) | 0.59 (0.42 to 0.84) | <0.01 | 1.29 (0.82 to 2.02) | 0.27 | 0.61 (0.43 to 0.87) | <0.01 | 1.26 (0.80 to 1.99) | 0.33 |

Significant p values (p<0.05) are printed in italic.
Model 1: adjusted for age; model 2: adjusted for age, hypertension, hypercholesterolemia, diabetes and smoking status.
*Age-adjusted prevalence.
†Additionally adjusted for ethnicity.
‡Measure of effect modification on multiplicative scale (statistical interaction term).
NA, not applicable; SA, South-Asian.

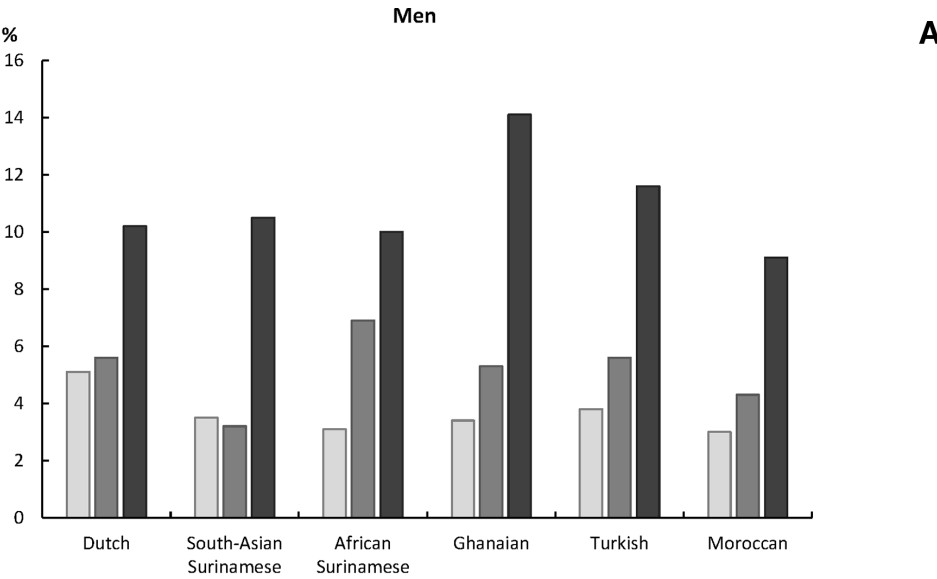

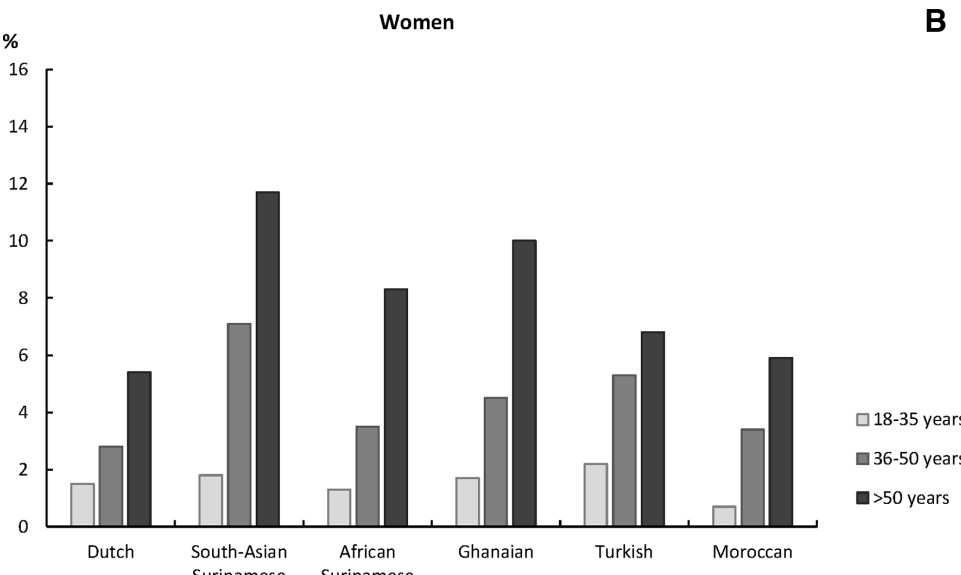

**Figure 1** Prevalence of any major ECG abnormalities in men (A) and women (B) by age-groups and ethnicity.

cardiomyopathy (1.1%) among women and T-wave abnormalities (1.6%), right bundle branch block (RBBB; 1.4%) and cardiomyopathy (1.1%) among men. Among South-Asian Surinamese women, the only group of women with no lower odds than men, microvoltages (2.9%), cardiomyopathy (1.9%) and T-wave abnormalities (1.5%) were the most prevalent major ECG abnormalities. T-wave abnormalities (1.9%), cardiomyopathy (1.2%) and RBBB (1.1%) were the most prevalent major ECG abnormalities among South-Asian Surinamese men.

As expected, the prevalence of major ECG abnormalities was higher in older than younger age-groups in both men and women (figure 1). The general pattern of smaller sex differences in the ethnic minority groups compared with the Dutch differed across the age strata (table 3). In the older age-groups, the adjusted sex difference in the odds of having a major ECG abnormality appeared less pronounced in ethnic minorities compared with the

Dutch, whereas this sex difference appeared more similar across ethnic group in the youngest age-group. Whereas women in all ethnic groups had a lower odds compared with men across all age strata, this was only the case in the youngest age-group of South-Asian Surinamese women vs men.

Women had a lower prevalence of minor ECG abnormalities (range: 16.2%–35.6%) compared with men (range: 28.5%–55.7%; online supplementary appendix table 3). Sex differences in minor ECG abnormalities were similar across ethnic groups, and were not influenced by conventional risk factors.

The prevalence of most common ECG abnormalities was also lower in women than men (online supplementary appendix table 4A). Only mildly prolonged QTc interval was more prevalent in women than in men. Patterns were similar across ethnic groups. The prevalence of most less common ECG abnormalities was also lower in women

**Table 3** The odds of major ECG abnormalities in women compared with men by age-groups, in the total population and with an interaction term for sex and ethnicity

| | OR (95% CI) | P value | Ratio of ORs (95% CI)‡ | P value |
|---|---|---|---|---|
| Aged 18–35 years (n=5870)* | | | | |
| Overall† | 0.38 (0.27 to 0.54) | <0.001 | NA | NA |
| Dutch | 0.30 (0.14 to 0.60) | <0.01 | Reference | NA |
| South-Asian Surinamese | 0.48 (0.18 to 1.18) | 0.12 | 1.58 (0.47 to 5.11) | 0.45 |
| African Surinamese | 0.38 (0.12 to 1.05) | 0.07 | 1.25 (0.34 to 4.44) | 0.73 |
| Ghanaian | 0.48 (0.13 to 1.77) | 0.26 | 1.60 (0.37 to 7.07) | 0.52 |
| Turkish | 0.55 (0.28 to 1.09) | 0.09 | 1.84 (0.69 to 5.01) | 0.23 |
| Moroccan | 0.23 (0.09 to 0.55) | <0.01 | 0.77 (0.23 to 2.42) | 0.65 |
| Aged 36–50 years (n=7099)* | | | | |
| Overall† | 0.89 (0.70 to 1.12) | 0.32 | NA | NA |
| Dutch | 0.51 (0.28 to 0.90) | 0.02 | Reference | NA |
| South-Asian Surinamese | 2.61 (1.39 to 5.20) | <0.01 | 5.13 (2.19 to 12.56) | <0.001 |
| African Surinamese | 0.49 (0.29 to 0.85) | 0.01 | 0.97 (0.44 to 2.16) | 0.94 |
| Ghanaian | 0.91 (0.50 to 1.71) | 0.76 | 1.79 (0.78 to 4.22) | 0.18 |
| Turkish | 1.04 (0.64 to 1.69) | 0.87 | 2.04 (0.97 to 4.38) | 0.06 |
| Moroccan | 0.92 (0.51 to 1.67) | 0.79 | 1.81 (0.80 to 4.17) | 0.16 |
| Aged >50 years (n=6489)* | | | | |
| Overall† | 0.74 (0.62 to 0.89) | <0.01 | NA | NA |
| Dutch | 0.54 (0.37 to 0.78) | <0.01 | Reference | NA |
| South-Asian Surinamese | 1.22 (0.79 to 1.91) | 0.37 | 2.27 (1.28 to 4.06) | <0.01 |
| African Surinamese | 0.84 (0.59 to 1.18) | 0.31 | 1.55 (0.94 to 2.59) | 0.09 |
| Ghanaian | 0.74 (0.46 to 1.17) | 0.20 | 1.37 (0.75 to 2.48) | 0.30 |
| Turkish | 0.55 (0.31 to 0.96) | 0.04 | 1.02 (0.51 to 2.00) | 0.96 |
| Moroccan | 0.68 (0.40 to 1.16) | 0.16 | 1.26 (0.66 to 2.42) | 0.49 |

*Model adjustment: age, hypertension, hypercholesterolemia, diabetes and smoking status.
†These models were also adjusted for ethnicity.
‡Measure of effect modification on multiplicative scale (statistical interaction term).
NA, not applicable.

than men, except for microvoltages, severely prolonged QTc (Bazett) interval, and left bundle branch block, atrial rhythm and sinus tachycardia (online supplementary appendix table 4B).

Sensitivity analyses did not alter our interpretation of findings (data not shown).

## DISCUSSION

In our study, women have an overall lower age-adjusted prevalence of major ECG abnormalities than men. Sex differences in the prevalence of major ECG abnormalities are smaller in the ethnic minority groups than in the Dutch, particularly in older age-groups. Differences in conventional cardiovascular risk factors and other well-known risk factors do not contribute substantially to these sex differences.

Our study has limitations. First, the results may be affected by selection bias due to non-response (response

rate: 28%). Non-response analyses showed that women were more likely to participate than men, Turks and Moroccans were less likely to participate compared with other ethnic groups, and participants were slightly older than non-participants.[17] However, we were able to include large numbers of both men and women, each ethnic group and age-group, indicating sufficient representation of all subgroups. This is relevant because previous work has shown that relative differences in CVD risk factors between ethnic groups are similar to other European countries,[28] suggesting that our results are generalisable to other European countries. Second, the definition of prior CVD was not comprehensive, as data on self-reported prior CVD other than myocardial infarction and CVA were lacking. However, we also excluded participants with a prior angioplasty or bypass surgery (on heart or legs), or paced rhythms, and those participants using antiplatelet drugs, oral

anticoagulants or antiarrhythmic agents and verified that our results were consistent in analyses restricted to those with a favourable cardiovascular risk profile (post hoc analysis in participants without hypertension, hypercholesterolemia, diabetes and those not smoking; data not shown). Therefore, it is unlikely that our results were substantially affected by misclassification. Third, single ECG measurements may have been suboptimal for the measurement of ECG abnormalities, potentially affecting the prevalence estimates. Some common expressions of CVD might not always be detectable by a single ECG measurement of 10 s, such as paroxysmal atrial fibrillation, and some ECG abnormalities need additional diagnostic measurements. However, 24 hours ECG monitoring with portable ECG devices and additional imaging techniques (eg, echocardiography) are often not feasible in population-based studies. Finally, the classification of 'major' and 'minor' ECG abnormalities depends on criteria used, and may be variable given the complexity of detailed ECG interpretation. For instance, the level of severity of some abnormalities may depend on the full clinical assessment or on combinations of abnormalities (eg, RBBB with left axis deviation). We also did not distinguish between men and women, ethnic groups, and age-groups in the assessment of the classification of major and minor abnormalities. If future research would reveal that the implication of abnormalities is different for any of these groups, this may influence the magnitude of the observed differences in our study.

Similar to previous studies reporting on the prevalence of composite major ECG abnormalities stratified by sex,[5–8 10 11] we observed an overall lower prevalence of major ECG abnormalities in women compared with men in most ethnic groups. Prevalence estimates in both men and women were within the range reported in most previous studies (range: 3.0%–13.2%),[5 6 8 10] except two studies with higher estimates.[7 11] T-wave abnormalities were the most prevalent major ECG abnormalities in both men and women in our study and most previous studies.[5–7 10 11] A much larger heterogeneity has been reported in previous studies in prevalence of minor ECG abnormalities, ranging from 4.5% to 31.6% in women[5 6 8 9 11] and from 7.3% to 45.7% in men.[5 6 8 9 11] Our prevalence was higher compared with most studies, most likely due to differences in the classification of major and minor ECG abnormalities.

The observed sex differences in major ECG abnormalities are in line with known differences in cardiovascular pathophysiology and epidemiology of CVD between men and women.[29 30] For instance, men tend to develop coronary artery disease (CAD) earlier than women, resulting in a higher incidence of CAD in men compared with women, in particular at a younger age.[29 30] This age-effect is consistent with our observations across ethnic groups of larger sex differences in prevalence of major ECG abnormalities in the youngest age-group compared with the older age-groups.

Differential patterning of cardiovascular risk factors did not explain the observed sex differences in prevalence of major ECG abnormalities overall and across ethnic groups. This finding is consistent with two previous studies on sex differences in ECG abnormalities[5 6] but not with another study.[7] Other explanations for the relative cardiovascular advantage of women compared with men at a younger age are still unclear, but may relate to sex hormones, with a prominent role for the protective effects of oestrogen in the development of CVD among premenopausal women.[29] Our findings of larger sex differences in prevalence of major ECG abnormalities in the youngest age-group compared with the older age-groups support this hypothesis.

We observed that only South-Asian Surinamese women did not have a lower odds of having a major ECG abnormality compared with South-Asian Surinamese men, which was mainly due to the higher prevalence of major ECG abnormalities among South-Asian Surinamese women compared with other women. South-Asian populations living in Europe are already considered a high-risk population for CVD[14] and our findings may suggest that South-Asian Surinamese women specifically are a target group for CVD prevention strategies. Although women had a consistently lower odds of having a major ECG abnormality than men in all other ethnic groups (except South-Asian Surinamese), Dutch women had a larger cardiovascular advantage than the other women. These findings are in line with a previous study from the USA showing a larger gap between men and women of the white majority population compared with black men and women in CAD mortality.[31] In contrast, a Dutch study on sex disparities in myocardial infarction incidence observed a smaller sex difference in the Dutch majority population compared with minority populations originating from Morocco, South-Asia and Turkey.[15] Explanations for the discrepancy between this and our study are unclear, but may relate to differences in study populations and exclusion criteria.

Differential patterning of cardiovascular risk factors did not explain the smaller cardiovascular advantage among minority women compared with Dutch women, suggesting that other factors may be relevant. Psychosocial factors (eg, discrimination), for instance, may be important risk factors for major ECG abnormalities in some groups of participants, potentially through stress and lifestyle-related factors. For instance, an American study found that current and chronic stress were associated with subclinical atherosclerosis in South-Asian women but not in South-Asian men.[32] Further research needs to confirm whether these psychosocial factors may also explain ethnic-specific variation in sex differences in occurrence of major ECG abnormalities.

The observed sex differences in occurrence of major ECG abnormalities, overall and within subgroups, may also reflect that ECG reference values do not differentiate between men and women (except QTc duration), ethnic groups or age-groups. Normal values for ECGs may differ

for women[33] and non-white groups[34] compared with white men, in whom the ECG reference criteria were developed. This is problematic since subgroups with pathological ECGs and potentially related cardiovascular risk might have been missed, or have a false positive diagnosis. For example, some studies suggest that current ECG criteria for microvoltages may be less valid for women[35] and Asian populations,[36] which may have resulted in an overestimation of the occurrence of microvoltages among South-Asian Surinamese women in our study.

In conclusion, we observed sex differences in ECG abnormalities and identified subpopulations with a relatively high prevalence, for example, Dutch men, and men and women of South-Asian and African origin. Given the association of major ECG abnormalities with CVD morbidity and mortality,[2–4] these groups may particularly benefit from prevention strategies to reduce the future burden of CVD. Moreover, the observed differences occurred irrespective of conventional risk factors, suggesting that opportunities to reduce the burden of CVD might be missed if prevention strategies are solely targeted at those with conventional risk factors. Previous studies have suggested that ECG measures may be, in addition to established cardiovascular risk factors, useful for the prediction of future CVD in intermediate and high-risk groups.[37 38] However, evidence is still limited, potential harms of screening are unknown, and ECG reference values are not sex-specific, ethnic-specific and age-specific. Hence, screening for CVD with ECG is currently not recommended.[39] In future research, ECG reference values should be validated in ethnically diverse populations of men and women of different age-groups in order to further investigate the occurrence of ECG abnormalities and the potentially added value of an ECG to cardiovascular risk classification.

**Author affiliations**
[1]Department of Public and Occupational Health, Amsterdam UMC, University of Amsterdam, Meibergdreef 9, Amsterdam, The Netherlands
[2]Department of Cardiology, Amsterdam UMC, University of Amsterdam, Meibergdreef 9, Amsterdam, The Netherlands
[3]Department of Cardiology, Leiden University Medical Center, Leiden, The Netherlands
[4]Department of General Practice, Amsterdam UMC, University of Amsterdam, Meibergdreef 9, Amsterdam, The Netherlands
[5]Netherlands Heart Institute, Utrecht, The Netherlands
[6]Department of Medical Informatics, Erasmus MC, University Medical Center Rotterdam, Rotterdam, The Netherlands
[7]Department of Clinical and Experimental Cardiology, Amsterdam UMC, University of Amsterdam, Meibergdreef 9, Amsterdam, The Netherlands
[8]Department of Clinical Epidemiology, Biostatistics and Bioinformatics, Amsterdam UMC, University of Amsterdam, Meibergdreef 9, Amsterdam, The Netherlands

**Acknowledgements** We are most grateful to the participants of the HELIUS study and the management team, research nurses, interviewers, research assistants and other staff who have taken part in gathering the data of this study.

**Contributors** RB and IGMvV contributed to the conception and design of the work. CCtH, REH, PGP and AEK contributed to the design. MBS and RJGP contributed to the acquisition of the data. All authors contributed to the analysis and interpretation of the results. RB drafted the manuscript. CCtH, REH, HLT, JAK, PGP, MBS, RJGP, AEK and IGMvV critically revised the manuscript. All authors gave final approval and agree to be accountable for all aspects of work ensuring integrity and accuracy.

**Funding** The HELIUS study is conducted by the Academic Medical Center Amsterdam and the Public Health Service of Amsterdam. Both organisations provided core support for HELIUS. The HELIUS study is also funded by the Dutch Heart Foundation, the Netherlands Organization for Health Research and Development (ZonMw), European Union (FP-7) and the European Fund for the Integration of non-EU immigrants (EIF). This work was additionally supported by a grant from the ZonMw Gender and Health Program (grant number 849200008). HLT has received funding from the European Union's Horizon 2020 research and innovation programme under acronym ESCAPE-NET (grant number 733381).

**Competing interests** None declared.

**Patient consent for publication** Not required.

**Provenance and peer review** Not commissioned; externally peer reviewed.

**Data availability statement** Data are available upon reasonable request.

**ORCID iD**
Renee Bolijn http://orcid.org/0000-0002-0803-6118

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
