## [Reviewer comments · BMJ Open]

ARTICLE DETAILS

TITLE (PROVISIONAL)	Do sex differences in the prevalence of electrocardiographic abnormalities vary across ethnic groups living in the Netherlands? A cross-sectional analysis of the population-based HELIUS study
AUTHORS	Bolijn, Renee; ter Haar, Cato; Harskamp, Ralf; Tan, Hanno; Kors, Jan; Postema, Pieter; Snijder, Marieke; Peters, Ron; Kunst, Anton; van Valkengoed, Irene

VERSION 1 – REVIEW

REVIEWER	Zehuai Wen Guangdong Provincial Hospital of Chinese Medicine , Guangzhou University of Chinese Medicine
REVIEW RETURNED	23-Apr-2020

GENERAL COMMENTS	The study used the data from the HELIUS study to explore sex differences in the prevalence of ECG abnormalities in adults living in Amsterdam. This is an interesting survey in this field to answer a gap question. The report of the study followed the principles of the STROBE statement. Only minor comments are below. 1.It is best to have a brief introduction to stratification random sampling method described in the manuscript.2.For the statistical analyses section, the dependent variable is major or minor ECG abnormalities, does it consider a multinomial logistic regression with a three-categories dependent variable of ECG abnormalities?
---

REVIEWER	Professor Mary Ward Ulster University Cromore Rd Coleraine BT52 1SA
REVIEW RETURNED	21-May-2020

GENERAL COMMENTS	This is a well written manuscript that investigates sex differences in the prevalence of ECG abnormalities across ethnic groups in the HELIUS cohort. The cross sectional study includes data on almost 20,000 adults from this multi-ethnic, population based study and reports that women have a lower prevalence of major ECG abnormalities than men, which is consistent with previous literature in the field. This is one of a many studies published on this cohort by the research group and presents new data to add to the evidence base, particularly regarding the differences by ethnic group. Strengths and limitations have been identified, including the size of the cohort which facilitates meaningful sub-analysis. The paper is well written and the abstract is clear although care should be taken not to over extrapolate the conclusion given the
---

	modest differences observed in major ECG abnormalities – suggest that this is amended. The Introduction sets the scene and provides a clear rationale however there appear to be additional studies that have considered gender / sex differences from a quick search of the literature that have not been referenced in this manuscript therefore the statements that there are only should be modified / clarifies eg Ethnic and Gender Specific Differences Among Athletes Participating in ECG Screening - American College of Cardiology https://www.acc.org/latest-in-cardiology/articles/2016/08/25/12/25/ethnic-and-gender-specific-differences-among-athletes-participating-in-ecg-screening#.XsZbRfiOhIs.twitter Methods: It would be useful to see a study flow diagram in the manuscript or as an appendix if space is limited to show account for numbers included in the analysis. Discussion: caution re over-interpretation of the findings. Query the rational for the structure of the discussion section i.e. placing strengths and limitations after the opening paragraph however this is an editorial decision. The authors highlight the issue of ‘classification’ of major and minor – this is a significant issue given the relatively small percentages reported and modest difference between groups, are the authors expand the discussion section to give further consideration to this limitation. The implications of the study have been highlighted however the authors have not included a clear conclusion based on the findings reported. Strobe guidelines considered.
--	--

VERSION 1 – AUTHOR RESPONSE

Reviewer: 1

Reviewer Name: Zehuai Wen

Institution and Country: Guangdong Provincial Hospital of Chinese Medicine, Guangzhou University of Chinese Medicine

Please state any competing interests or state ‘None declared’: None declared.

Please leave your comments for the authors below

The study used the data from the HELIUS study to explore sex differences in the prevalence of ECG abnormalities in adults living in Amsterdam. This is an interesting survey in this field to answer a gap question. The report of the study followed the principles of the STROBE statement. Only minor comments are below.

REPLY: We thank the reviewer for acknowledging the interest of the topic.

1.It is best to have a brief introduction to stratification random sampling method described in the manuscript.

REPLY: We have clarified the random sampling method in the method section. In addition, we included an additional reference to the design paper of the HELIUS study to provide more information on development of the cohort:

‘The HELIUS study has been described in detail elsewhere.[16,17]. Briefly, baseline data collection took place between 2011 and 2015 and included participants of Dutch, South-Asian Surinamese, African Surinamese, Ghanaian, Moroccan, and Turkish origin aged 18-70 years living in Amsterdam. Potential participants were sampled with a simple random sampling method from the municipality

registry, after stratification by ethnicity as defined by registered country of birth.[18]

2. For the statistical analyses section, the dependent variable is major or minor ECG abnormalities, does it consider a multinomial logistic regression with a three-categories dependent variable of ECG abnormalities?

REPLY: We have performed binary logistic regression analyses with any major ECG abnormalities (yes/no) and any minor ECG abnormalities (yes/no) as separate outcomes. We considered multinomial logistic regression not to be appropriate since major ECG abnormalities and minor ECG abnormalities are not mutually exclusive, i.e., they can co-occur.

We clarified this in the method section: 'We performed binary logistic regression analyses with hierarchal models to examine sex differences in prevalence of 1) any major ECG abnormalities and 2) any minor ECG abnormalities, adjusted for ...'

Reviewer: 2

Reviewer Name: Professor Mary Ward

Institution and Country: Ulster University, Cromore Rd, Coleraine, BT52 1SA

Please state any competing interests or state 'None declared': None declared

Please leave your comments for the authors below

This is a well written manuscript that investigates sex differences in the prevalence of ECG abnormalities across ethnic groups in the HELIUS cohort. The cross sectional study includes data on almost 20,000 adults from this multi-ethnic, population based study and reports that women have a lower prevalence of major ECG abnormalities than men, which is consistent with previous literature in the field. This is one of a many studies published on this cohort by the research group and presents new data to add to the evidence base, particularly regarding the differences by ethnic group. Strengths and limitations have been identified, including the size of the cohort which facilitates meaningful sub-analysis.

1. The paper is well written and the abstract is clear although care should be taken not to over extrapolate the conclusion given the modest differences observed in major ECG abnormalities – suggest that this is amended.

REPLY: We thank the reviewer for acknowledging the interest of the topic. We describe the prevalence of ECG abnormalities in a healthy, general population sample (from which people with known CVD were excluded). Although the differences may appear modest in absolute terms, we consider the relative differences relevant as they translate to substantial numbers at a population level. Nevertheless, we have reformulated our conclusion: 'The prevalence of major ECG abnormalities was lower in women than men. However, sex differences were less apparent in ethnic minority groups. Conventional risk factors did not contribute substantially to observed sex differences.'

2. The Introduction sets the scene and provides a clear rationale however there appear to be additional studies that have considered gender / sex differences from a quick search of the literature that have not been referenced in this manuscript therefore the statements that there are only should be modified / clarifies eg Ethnic and Gender Specific Differences Among Athletes Participating in ECG Screening - American College of Cardiology

<https://eur04.safelinks.protection.outlook.com/?url=https%3A%2F%2Fwww.acc.org%2Flatest-in-cardiology%2Farticles%2F2016%2F08%2F25%2F12%2F25%2Fethnic-and-gender-specific-differences-among-athletes-participating-in-ecg-screening%23.XsZbRfiOhIs.twitter&data=02%7C01%7Cr.bolijn%40amsterdamumc.nl%7C6d66457c5d3b46cb78fa08d803ebd8d8%7C68dfab1a11bb4cc6beb528d756984fb6%7C0%7C0%7C637263658780694462&sdata=p0ee2N%2B6%2Blx4L9Z0lns45qGWR90Mzcf66zGQEBKt8Cc%3D&reserved=0>

REPLY: We thank the reviewer for this suggestion. Although this is an interesting article, it focusses on ECG criteria for preparticipation screening in athletes. Athletes represent a selective subgroup, as

they are generally younger and healthier than the general population. Therefore, different ECG criteria may be applicable and another classification of abnormalities may be relevant. Since our study focusses on differences in ECG abnormalities in the general population, we consider this paper to be less applicable. We have clarified in the introduction section that the focus of our study is on sex differences in ECG abnormalities in the general population:

'Although studies have described the prevalence of major ECG abnormalities in men and women from diverse general populations,[5-11] only three studied sex differences in general populations specifically.[5-7]'

To be certain that we indeed did not miss general population studies, we have repeated our literature search and rechecked the reference lists of included papers for potentially relevant papers on sex differences in ECG abnormalities. However, this did not yield additional papers to be included in our manuscript.

3. Methods: It would be useful to see a study flow diagram in the manuscript or as an appendix if space is limited to show account for numbers included in the analysis.

REPLY: We have included a flow diagram in the appendix.

4. Discussion: caution re over-interpretation of the findings.

Query the rationale for the structure of the discussion section i.e. placing strengths and limitations after the opening paragraph however this is an editorial decision.

REPLY: We consider the presentation of the strengths and limitations before the interpretation of the results in the discussion section to be more insightful to the reader, as the reader is provided with all background information that is needed to value the discussion of our findings on beforehand.

Therefore, we would prefer to retain the current order of the discussion section.

5. The authors highlight the issue of 'classification' of major and minor – this is a significant issue given the relatively small percentages reported and modest difference between groups, are the authors expand the discussion section to give further consideration to this limitation.

REPLY: We have clarified the procedure of the classification of major and minor ECG abnormalities in the method section: 'We classified ECG abnormalities into major and minor ECG abnormalities, based on previous research[7] and consensus discussion among experts (Appendix Table 1). This classification was completed prior to data analysis.'

In addition, we expanded the discussion of this issue in the discussion section: 'Finally, the classification of 'major' and 'minor' ECG abnormalities depends on criteria used, and may be variable given the complexity of detailed ECG interpretation. For instance, the level of severity of some abnormalities may depend on the full clinical assessment or on combinations of abnormalities (e.g., RBBB with left axis deviation). We also did not distinguish between men and women, ethnic groups, and age-groups in the assessment of the classification of major and minor abnormalities. If future research would reveal that the implication of abnormalities is different for any of these groups, this may influence the magnitude of the observed differences in our study.'

6. The implications of the study have been highlighted however the authors have not included a clear conclusion based on the findings reported.

REPLY: We have changed the last section of the discussion to further emphasize the conclusion of our study, in light of potential implications: 'In conclusion, we observed sex differences in ECG abnormalities and identified subpopulations with a relatively high prevalence, e.g., Dutch men, and men and women of South-Asian and African origin. Given the association of major ECG abnormalities with CVD morbidity and mortality,[2-4] these groups may particularly benefit from prevention strategies to reduce the future burden of CVD. Moreover, the observed differences occurred irrespective of conventional risk factors, suggesting that opportunities to reduce the burden of CVD might be missed if prevention strategies are solely targeted at those with conventional risk factors. Previous studies have suggested that ECG measures may be, in addition to established

cardiovascular risk factors, useful for the prediction of future CVD in intermediate and high-risk groups.[37,38] However, evidence is still limited, potential harms of screening are unknown, and ECG reference values are not sex-, ethnic-, and age-specific. Hence, screening for CVD with ECG is currently not recommended.[39] In future research, ECG reference values should be validated in ethnically diverse populations of men and women of different age-groups in order to further investigate the occurrence of ECG abnormalities and the potentially added value of an ECG to cardiovascular risk classification.'

VERSION 2 – REVIEW

REVIEWER	Zehuai Wen Guangdong Provincial Hospital of Chinese Medicine, Guangzhou University of Chinese Medicine
REVIEW RETURNED	28-Jun-2020

GENERAL COMMENTS	Authors have appropriately responded to all my comments I made in the previous revision.
--

REVIEWER	Mary Ward Ulster University Northern Ireland
REVIEW RETURNED	16-Jul-2020

GENERAL COMMENTS	The authors have defended the position, within the manuscript, of the strengths and limitations section. This is an editorial matter. Otherwise I am happy that my comments have been adequately addressed.
---